# Permeable Water-Resistant Heat Insulation Panel Based on Recycled Materials and Its Physical and Mechanical Properties

**DOI:** 10.3390/molecules24183300

**Published:** 2019-09-11

**Authors:** Štěpán Hýsek, Miroslav Frydrych, Miroslav Herclík, Ludmila Fridrichová, Petr Louda, Roman Knížek, Su Le Van, Hiep Le Chi

**Affiliations:** 1Department of Material Science, Faculty of Mechanical Engineering, Technical University of Liberec, Studentska 2, 461 17 Liberec, Czech Republic; 2Department of Textile Evaluation, Faculty of Textile Engineering, Technical University of Liberec, Studentska 2, 461 17 Liberec, Czech Republic,

**Keywords:** heat insulation, sandwich panel, polyurethane foam, geopolymer, nanofiber membrane

## Abstract

This paper deals with the development and characteristics of the properties of a permeable water-resistant heat insulation panel based on recycled materials. The insulation panel consists of a thermal insulation core of recycled soft polyurethane foam and winter wheat husk, a layer of geopolymer that gives the entire sandwich composite strength and fire resistance, and a nanofibrous membrane that permits water vapor permeability, but not water in liquid form. The observed properties are the thermal conductivity coefficient, volumetric heat capacity, fire resistance, resistance to long-term exposure of a water column, and the tensile strength perpendicular to the plane of the board. The results showed that while the addition of husk to the thermal insulation core does not significantly impair its thermal insulation properties, the tensile strength perpendicular to the plane of these boards was impaired by the addition of husk. The geopolymer layer increased the fire resistance of the panel for up to 13 min, and the implementation of the nanofibrous membrane resulted in a water flow of 154 cm^2^ in the amount of 486 g of water per 24 h at a water column height of 0.8 m.

## 1. Introduction

One of the most important challenges for the construction industry is to reduce the energy demands of buildings throughout their entire life cycle. During the use of a building, its thermal demands are undoubtedly influenced by its insulation. Commonly used thermal insulation materials for building insulation are produced from petrochemical products or from natural sources, but their production is highly energy intensive (glass, rock, wool) [1]. From this perspective, the use of recycled and plant materials is very promising for the production of thermal insulation. In the case of plant materials, rice husks [2], sunflower stalks [3], wheat straw [4], wheat husks [5] flax fibers [6], hemp fibers [7], larch bark [8], and many others can be considered for thermal insulation production. Recycling synthetic materials or using agricultural or industrial residues can be an effective way to reduce virgin materials consumption [9]. Products from recycled plastics such as polyethylene terephthalate [10] and recycled textile fibers [11] provide very good thermal insulation properties.

However, a significant disadvantage of plant materials consisting mainly of cellulose, hemicelluloses, lignin, and pectins is their flammability [12]. In terms of building materials, their resistance to burning by geopolymer applications [13,14] can be improved significantly. Geopolymers are materials usually synthesized using an aluminosilicate raw material and an activating solution mainly composed of alkalis of sodium or potassium and water glass [15,16]. Due to the properties of geopolymers in the form of high strength, resistance to chemicals and, in particular, thermal stability and fire resistance, applications of these geopolymers in the form of protective coatings or coatings on structures [17,18,19,20,21] have been studied in recent years. In the past, the fire resistance of particleboards based on winter rapeseed stalks [13] has been successfully increased by the geopolymer layer. Even better geopolymer properties can be achieved, for example, via the implementation of carbon fibers, which result in better mechanical properties of the entire composite [22].

An important property of cellulose-based plant fibers is hygroscopicity. This property may be an advantage in some applications and a disadvantage in other applications. However, in terms of thermal insulation of structures, high humidity in the insulation is undesirable, as water reduces the thermal insulation properties of the material [23]. On the other hand, we require, from natural fiber, thermal insulation interior vapor permeability through the building envelope to the exterior [24,25]. Preventing the permeability of liquid water from the exterior into the building envelope and, at the same time ensuring the transport of water vapor from the interior through the building envelope to the exterior, is ensured by a suitably-selected wall structure [26]. One of the elements that can be used in the wall structure for this purpose can be a nanofibrous membrane, which provides water vapor permeability, but prevents the permeability of water in the liquid state [27]. In addition, a suitably-designed nanofibrous membrane can withstand a very high water column, which can affect the building, for example during floods [27].

This paper deals with the use of post-harvest residues of winter wheat and recycled polyurethane foam in combination with geopolymer foam and a nanofibrous membrane for the production of composite materials with properties for the given purpose of use. The aim of this work is to determine the influence of winter wheat husk and the implementation of a nanofibrous membrane and a geopolymer layer into the sandwich panel structure on its mechanical and physical properties. This paper contributes to finding material utilization of wheat husks, which provides storage of CO_2_ in comparison with energetic utilization of this raw material. Moreover, addition of husks into the heat insulation panel may bring additional benefits during manufacturing of these panels. Since wheat husks contain 12.7% moisture [5], no steam injection would be necessary for hardening of polyurethane adhesive.

## 2. Materials and Methods

### 2.1. Heat Insulation Board Manufacturing

The insulation boards were made of crushed flexible polyurethane (PUR) foam, winter wheat husk, and PU4349/3 one component moisture curing binder (Leeson Polyurethanes Ltd., Warwick, UK). The crushed flexible PUR foam was supplied by the Molitan company (Molitan a. s., Breclav, Czech republic) as recyclate from manufacturing rests. The apparent density of the used PUR foam was 24 kg/m^3^ and the bulk density was 11.3 kg/m^3^. The PUR particle fraction analysis is shown in the results. Winter wheat husks were mixed into the boards at 0% and 25% to the weight of the PUR recycled material. The analysis of the husk fraction is presented in the results. The adhesive was applied to the particles by spraying in a laboratory adhesive applicator, and the proportion of adhesive on the dry matter was 15%. The carpet was manually layered and compressed between the steel screens, and curing was carried out in a heat chamber at an air temperature of 120 °C for 15 min. The boards were then air conditioned at 20 °C and 65% relative humidity (RH) for 3 days. Figure 1 shows the surface view of the thermal insulation boards.

### 2.2. Geopolymer and Nanofiber Membrane Application

A geopolymer layer of 1 cm thickness and a density of 880 kg/m^3^ was applied to one side of the insulation board to increase its fire resistance. The composition of the geopolymer is shown in Table 1. A more detailed identification of its composition is given in previously published research [13]. A nanofibrous membrane was manually deposited on the surface of the freshly-applied and uncured geopolymer. The nanofibrous membrane was implemented into the composite due to the above-described reason in order to prevent the permeability of water molecules in a liquid state, but allowing for the permeability of water vapor. In order to protect the nanofiber membrane from damage, it was laminated between two non-woven fabrics made from polyester with a basic weight of 55.6 g/m^2^. The nanofiber membrane was made of polyurethane via electrospinning using Nanospider technology (Elmarco s.r.o., Liberec, Czech Republic). The solution was spun in an electric field with a voltage of 80.7 kV, the distance of the condenser was 190 mm, the velocity of the supporting base fabric was 0.1 m/min, the relative humidity in the spinning chamber was 21%, and the surface weight of the produced nanofibrous layer was 6 g/m^2^.

Table 2 shows the variants of the sandwich composites being developed. Two variants of the percentage husk representation were chosen and composites with and without a membrane were made. The geopolymer layer was always constant. Figure 2 shows a cut of the sandwich panel.

### 2.3. Physical and Mechanical Properties Estimation

All of the tests were carried out after air conditioning of the material under conditions of 20 °C and 65% relative humidity. The distribution of husks and crushed PUR foam fraction was determined via a screen analysis and the results were then graphically expressed. The density of the material was determined according to standard EN 323 [28] and internal bonding (tensile strength perpendicular to the plane of the board) according to EN 319 [29]. The methodology of these experiments is described in more detail in [13]. The thermal insulation properties of boards were measured using the Isomet 2104 device (Applied Precision, Ltd., Bratislava, Slovakia) according to the method described previously in [30], using a probe with a measuring range of 0.015 to 2 W/(m·K). The thermal conductivity coefficient of the entire sandwich panel was determined by a calculation, because the thermal insulation properties of the sandwich materials cannot be measured by the used method. The calculation was carried out according to the thermal resistances of the individual layers (Equation (1)) and; therefore, the total thermal conductivity coefficient of the developed sandwich panels is a theoretical value that is based on the thermal resistance values of the individual layers and does not include thermal resistance during heat transfer.
(1)$$\lambda_{\mathrm{tot}}=\frac{d_{\mathrm{tot}}}{\sum R_{i}}=\frac{d_{\mathrm{tot}}}{\sum \frac{d_{i}}{\lambda_{i}}},$$
where $\lambda_{\mathrm{tot}}$. is the total thermal conductivity coefficient of the sandwich panel, $d_{\mathrm{tot}}$ is the total thickness of the sandwich panel, $d_{i}$. is the thickness of one layer in the sandwich panel, $\lambda_{i}$ is the thermal conductivity coefficient of one layer in the sandwich panel, and $R_{i}$ is the thermal resistance of one layer in the sandwich panel.

The fire resistance of the panels was performed via a thermal loading test. This test was performed according to the methodology previously published in [13], and comes from slightly modified standard EN 1363-2 [31]. A custom-designed furnace that allows for testing samples with dimensions of 300 mm × 300 mm was employed in order to characterize the behavior of the developed panels in different types of fire. Chosen external fire curves are presented in the results. Two temperature sensors were used, the first located in the burner chamber and the second on the outside of the flame. The course of temperatures was monitored over time. The flame intensity was controlled by the flow of gas and the flame was directed parallel to the plane of the tested sample.

The water permeability of sandwich composites was measured on our developed prototype. Unlike similar devices used to measure, for example, water column resistance, the used prototype measures the actual amount of liquid that the test sample releases over time at a defined hydrostatic pressure [27]. Samples with a circular cross section with a diameter of 17 cm were mounted in a test capsule using a seal and, subsequently, the surface of the sample of 154 cm^2^ was exposed to a water column 80 cm in height, corresponding to a pressure of 7.8 kPa. The water that passed through the composite was measured for 24 h. Throughout the experiment, the constant height of the water column to which the composite was exposed was maintained.

### 2.4. Statistical Analysis

Data was statistically processed using Statistica12 software (Tulsa, OK, USA). Descriptive statistics and graphical representations were used to describe the data. The influence of the observed factors on the variables was shown graphically: Thermal conductivity coefficient, thermal capacity, tensile strength perpendicular to the level of the board. The vertical columns correspond to 95% confidence intervals. Subsequently, a Tukey posthoc test was performed to determine if any of the differences between sample means were statistically significant. A significance level of α = 0.05 was used for all analyses. The temperature course during the thermal loading test was depicted using point chart as a function of time.

## 3. Results and Discussion

Figure 3 and Figure 4 show the distribution of the PUR foam crushed fraction and the winter wheat husk. While the predominant part of the crushed PUR foam particles is in the range of 5 to 15 mm, the predominant part of the husk can be characterized by dimensions of 1.5 to 3 mm. The particle size has a major influence on the mechanical properties of the boards [32]; however, in the case of the sandwich panels, where one layer is significantly stronger than the other, the geopolymer layer takes over all the flexural strength [33]. In this research, the particle size affected tensile strength perpendicular to the level of the board.

Figure 5 shows the effect of the weight ratio of husk in the insulation board on the thermal conductivity coefficient. The picture shows that in both cases the measured thermal insulation cores achieved very good thermal conductivity values in the range from 0.0427 to 0.0452 W/(m·K). The addition of the husk to the crushed PUR foam resulted in a slight deterioration of 0.0025 W/(m·K) (a statistically significant difference); nevertheless, these are still very good values compared to other alternative raw materials. The achieved thermal conductivity values are slightly lower than in the case of thermal insulation boards made from reeds [1], bagasse [34], or cotton stalks [35]. However, it should be noted that, in the above competing products, the manufactured boards had a higher density. For example, 30 kg/m^3^ recycled polyethylene terephthalate boards achieved a thermal conductivity coefficient of 0.0355 W/(m·K) [10].

Figure 6 shows the effect of the weight proportion of husk in the insulation board on the volumetric heat capacity. The difference between the individual variants is statistically significant at a level of 0.05. As with the thermal conductivity coefficient, the addition of husk increased this characteristic. However, in this case, this is an improvement in the characteristic that can compensate for the increase in the thermal conductivity coefficient, in the form of a higher accumulation capability of the material and the retention of heat in the walls at a slight decrease in exterior temperature [36]. However, panel cores containing husks achieved, still, a much lower volumetric heat capacity than another agriculture by-product—corn husks [37].

Table 3 shows the calculated thermal conductivity coefficient values of the entire sandwich composite panels and the measured density values of the individual materials. There were slight deviations in the actual thermal insulation board densities from their nominal values. The influence of nanofiber membranes on thermal insulation properties or fire resistance was not evaluated. The geopolymer layer only slightly worsened the thermal insulation properties of the sandwich composite. The total thermal conductivity coefficient is around 0.05 W/(m·K), which is a fully adequate value for thermal insulation materials [9], and produced panels are comparable to other commonly used materials [38]. The reached thermal conductivity coefficients are higher than the thermal conductivity coefficients of commercially-produced heat insulation panels from PUR or PIR (polyisocyanurate) panels; however, the developed panels are from recycled materials and from recycled PUR that was initially not produced for thermal insulation.

There was a statistically significant effect of the proportion of husk in the thermal insulation core on its internal bonding (Figure 7). With an increase in the proportion of husk in the material, internal bonding was reduced to 0.64 kPa, which is already insufficient for thermal insulation materials according to standard EN 13162+A1 [39]. For the production of industrially-useable thermal insulation panels with winter wheat husk admixtures, it would then be necessary either to select a higher proportion of adhesive [40] or to include pre-treatment of wheat husks in the production process, in order increase their surface energy and thus reach higher bonding [5].

The graphs in Figure 8 show the behavior of the entire panel under fire load. The samples were exposed to a flame with rapid (Figure 8a) and gradual (Figure 8b) temperature increases. No effect of the wheat husk additive on fire resistance was observed. However, the fire resistance of the boards was affected by the rate of temperature increase. In the case of a fast onset, the boards withstood the effect of flame for approximately 500 s, and more than 800 s in the case of gradual onset. Regardless of the steepness of the onset temperature curve, it was observed that when the temperature inside the furnace rises to around 400 °C, the temperature on the outer surface of thermal insulation boards increases to around 60 °C, which is then held constant until the material burns. These results correspond with results for sandwich-structured composites made from rapeseed stalks [13], and, because of the flammable insulation core, the panel withstood lower temperatures than in [16], where geopolymer composites were filled only by basalt microfibrils.

The developed sandwich panels were able to withstand fairly long-term exposure to a water column with a height of 80 cm. In 24 h, only 486 g of water flowed through the 154 cm^2^ area (Figure 9). There was no difference found between the sandwich panel with the addition of husk and no husk. All of the resistance of the sandwich composite to the long-term effect of the water column is due to the used nanofibrous membrane and the interface between the nanofibrous membrane and the geopolymer. With regard to the thermal insulation sandwich panel without a nanofibrous membrane, this sandwich is virtually unable to prevent water flow. When the sandwich without a nanofibrous membrane was encumbered with a water column with a height of 80 cm, 3700 g of water flowed through the 154 cm^2^ area over 4 min.

The results show that the geopolymer layer in the entire sandwich panel suitably complements the thermal insulation core. The geopolymer layer provided the material with fire resistance, and it can be assumed that it would increase flexural strength [41], while only slightly worsening the overall thermal conductivity coefficient. The geopolymer layer was thoroughly bonded to the thermal insulation core, and in the tensile strength test perpendicular to the plane of the board, there was no breach between these layers, but in the insulation core. The nanofibrous membrane also contributed to improving the properties of the entire sandwich composite. It gave the material resistance to long-term exposure to the water column, while not negatively affecting any other material properties.

## 4. Conclusions

The paper presented properties of a sandwich panel from recycled materials enhanced by a geopolymer layer and a nanofibrous membrane. It was shown that the addition of husk to the thermal insulation core increased the thermal conductivity coefficient up to the value of 0.0452 W/(m·K), but this negative increase can be compensated by the increase in specific heat capacity of the insulation core with husks up to the value of 0.126 MJ/(m^3^·K). The theoretical value of the thermal conductivity coefficient of the developed panels achieves excellent values on the level of 0.05 W/(m·K). The geopolymer layer and nanofibrous membrane provided the sandwich panel with the necessary properties for use as thermal insulation in exposed building walls, and fire resistance and water resistance increased nominally. The panel resisted fire with a gradual temperature increase for more than 13 min, and incorporation of a nanofibrous membrane provided enhanced resistance to a water column with a height of 0.8 m.

## Figures and Tables

**Figure 1 molecules-24-03300-f001:**
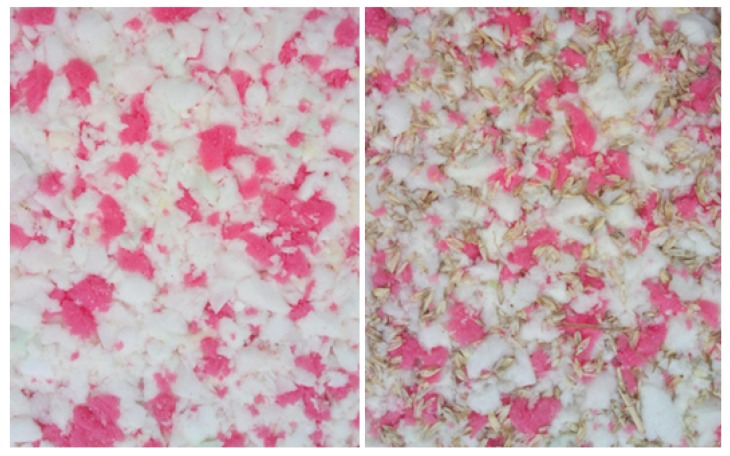
Surface view of the thermal insulation layer, board without husks on the **left**, board with addition of husks on the **right**.

**Figure 2 molecules-24-03300-f002:**
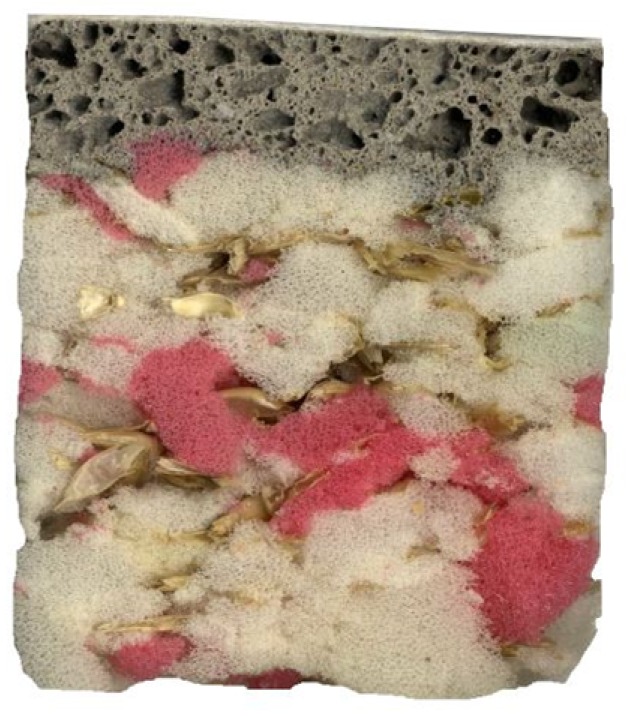
View of sandwich panel cut.

**Figure 3 molecules-24-03300-f003:**
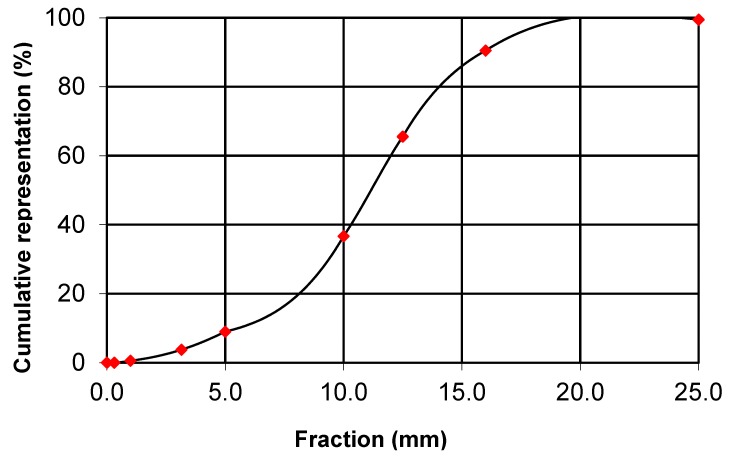
Fraction of crushed PUR foam.

**Figure 4 molecules-24-03300-f004:**
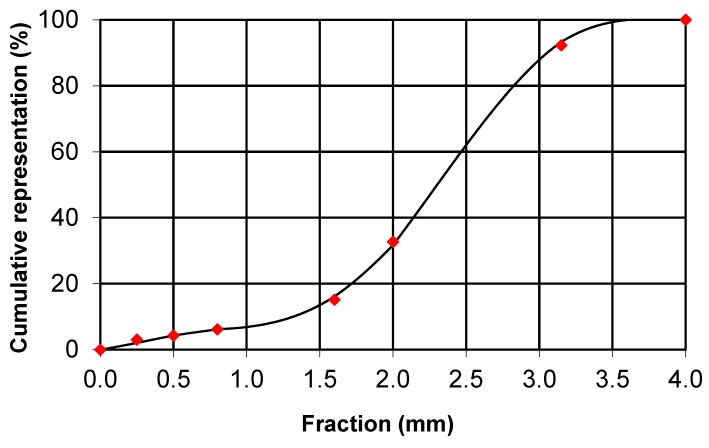
Fraction of winter wheat husk.

**Figure 5 molecules-24-03300-f005:**
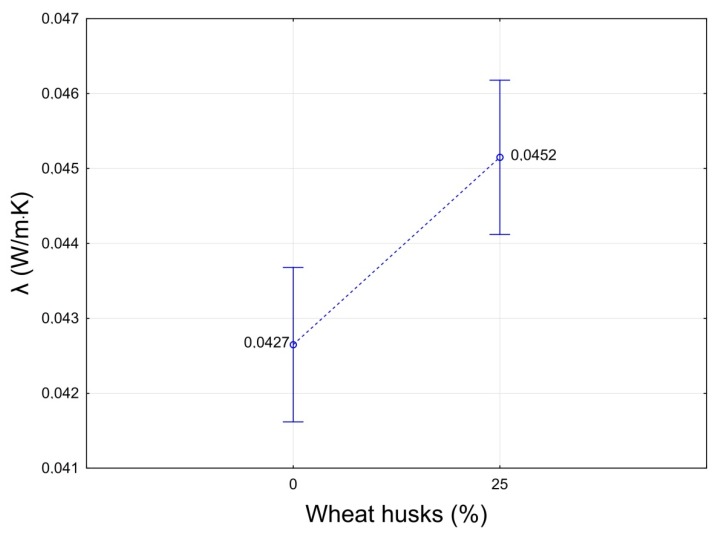
Influence of the proportion of the husk in the insulation board on the thermal conductivity coefficient.

**Figure 6 molecules-24-03300-f006:**
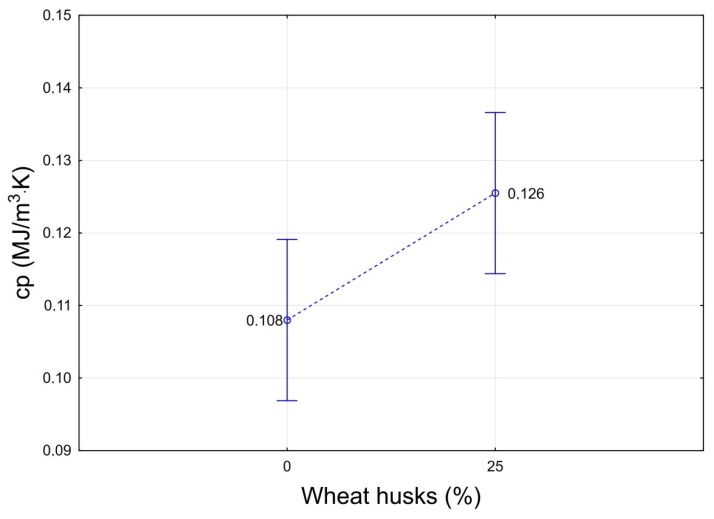
Influence of the proportion of the husk in the insulation board on volumetric heat capacity (cp).

**Figure 7 molecules-24-03300-f007:**
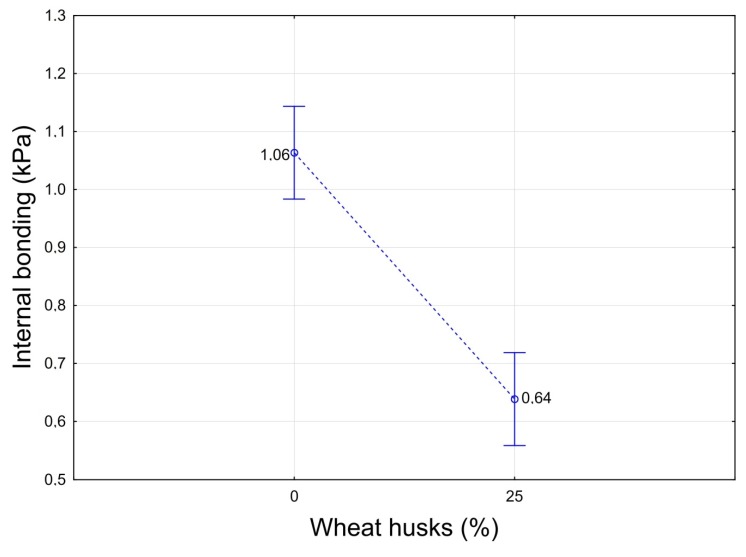
Influence of the proportion of the husk in the insulation board on internal bonding of composite materials.

**Figure 8 molecules-24-03300-f008:**
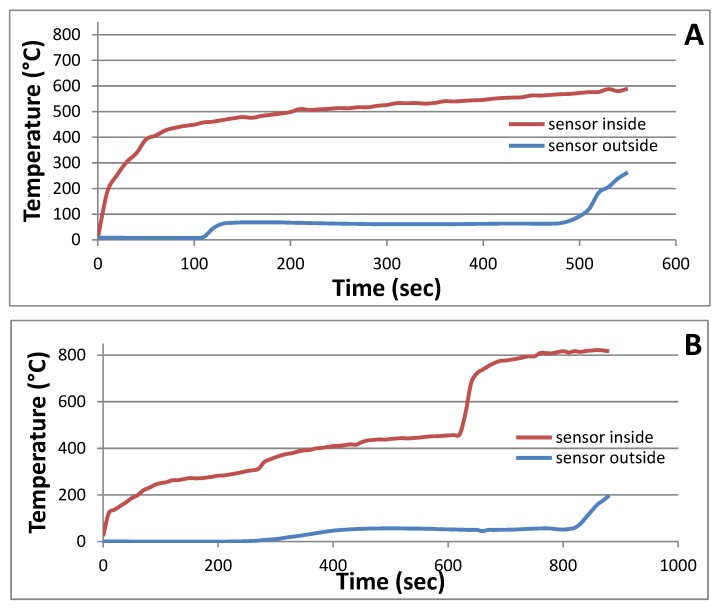
Burning characteristics of produced panels: (**A**) Rapid temperature increase; (**B**) gradual temperature increase.

**Figure 9 molecules-24-03300-f009:**
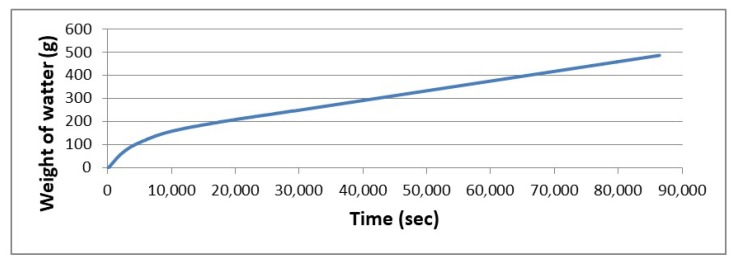
Effect of the nanofibrous membrane on the resistance of the sandwich panel against the long-term effects of the water column.

**Table 1 molecules-24-03300-t001:** Geopolymer composition.

Component	Percentage of Individual Components
Cement Baucis Lk	43.2%
Activator Baucis Lk	38.9%
KEMA MIKROSILIKA	4.3%
Mineral wool ISOVER	13.0%
Aluminum powder	0.6%

**Table 2 molecules-24-03300-t002:** Variants of the manufactured sandwich-structured panel.

	Permeable Water-Resistant Heat Insulation Panel			
**Recycled PUR:wheat husk ratio**	1:0		3:1	
**Nanofiber membrane**	0	1	0	1

Note: polyurethane (PUR).

**Table 3 molecules-24-03300-t003:** Average densities of materials and thermal conductivity of sandwich panels.

Recycled PUR:Wheat Husk Ratio	Heat Insulation Board Density (kg/m^3^)	Geopolymer Density (kg/m^3^)	λ20/65 (W/(m·K))
1:0	49.4 (1.7)	885 (32)	0.049 (0.006)
3:1	51.6 (4.2)	885 (32)	0.051 (0.006)

Note: Values in parentheses are the standard deviations. Polyurethane (PUR).

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
