# Peer review of "Permeable Water-Resistant Heat Insulation Panel Based on Recycled Materials and Its Physical and Mechanical Properties"

_molecules, 2019, doi:10.3390/molecules24183300_

Round 1

Reviewer 1 Report

Article entitled "Permeable Water-Resistant Heat Insulation Panel Based on Recycled Materials and its Physical and Mechanical Properties" is written very chaotically, which makes it very difficult to read. Article in this form is definitely not suitable for publication in Molecules journal. After the "Introduction" section, there should be a "Materials and Methods" section, not a "Results and Discussion" section. Article lacks transparency in this form. Reading this article to line 158 (end of "Result and Discussion" section), I had 12 elementary questions to which answers should be found in the text of this article. I found the answers to some of this questions in Chapter 3. This is confirmed by the fact that this chapter should be after the "Introduction" section. Regarding the missing information and ambiguities, I present my comments below:

- Authors wrote that they used "soft foam". I understand that it was flexible foam? Do authors know λ of the used foam ? I have doubts whether choosing such type of foam for thermal insulation purposes is right. Please explain your choose. In addition, authors should write more information about the used foam. From what and where the waste PUR foam was obtained. It is worth to include information about the blowing agent used to produce this foam. Foam fragments with a grain diameter of 10-30 mm may have a blowing agent closed inside (if it was a closed-cell foam). This has a huge impact on λ.

- Did the authors grind PUR foam themselves?

- How did authors measure the distribution of the crushed fraction?

- Value λ = 0.044 W/(m*K) is not a very good result.

- What standard did the authors use to conduct flammability tests for? Is this test adequate to the use of these materials in civil engineering? Earlier publication of this flammability testing methodology does not release authors from placing appropriate information in the text of the article.

-I suggest re-editing "Conclusion" section to a descriptive version. The most important numerical results (that has been improved) should be indicated there and the advantages of this solution should be emphasized.

Author Response

Dear reviewer,

thank you very much for your valuable suggestions. Please find enclosed to this cover letter the revised manuscript and responses to your comments.

I believe that after carefully performed revisions, the manuscript deserves to be considered for publication in the journal Molecules.

Best regards,

ŠtÄ›pán Hýsek

------------------------------

Response to Reviewer 1 Comments

Point 1: Article entitled "Permeable Water-Resistant Heat Insulation Panel Based on Recycled Materials and its Physical and Mechanical Properties" is written very chaotically, which makes it very difficult to read. Article in this form is definitely not suitable for publication in Molecules journal. After the "Introduction" section, there should be a "Materials and Methods" section, not a "Results and Discussion" section. Article lacks transparency in this form. Reading this article to line 158 (end of "Result and Discussion" section), I had 12 elementary questions to which answers should be found in the text of this article. I found the answers to some of this questions in Chapter 3. This is confirmed by the fact that this chapter should be after the "Introduction" section.

Response 1: "Materials and methods" section was moved before the “Results and discussion”. For better clarity, these two chapters were swapped first and then the revision tracking was turned on.

Point 2: - Authors wrote that they used "soft foam". I understand that it was flexible foam? Do authors know λ of the used foam ? I have doubts whether choosing such type of foam for thermal insulation purposes is right. Please explain your choose. In addition, authors should write more information about the used foam. From what and where the waste PUR foam was obtained. It is worth to include information about the blowing agent used to produce this foam. Foam fragments with a grain diameter of 10-30 mm may have a blowing agent closed inside (if it was a closed-cell foam). This has a huge impact on λ.

Response 2: More information about the used foam was added into the Materials and methods part and the term from soft to flexible was corrected. The crushed flexible foam was supplied by the Molitan company (Molitan a. s., Breclav, Czech republic) as recyclate from manufacturing rests. Unfortunately, it was not able to get more information and specifications of the foam. For the purpose of this study is important, that the foam used is recycled source. Its properties are given by supplier and in all experiments this material is constant.

Point 3: - Did the authors grind PUR foam themselves?

Response 3: No, the foam was supplied crushed.

Point 4: - How did authors measure the distribution of the crushed fraction?

Response 4: The fraction of husks and crushed PUR foam was determined via a sieve analysis. The sentence in the methodology part was reformulated.

Point 5: - Value λ = 0.044 W/(m*K) is not a very good result.

Response 5: It is generally understood that heat insulation materials for utilization as building insulation with λ < 0.05 W/(m.K) are good heat insulations and they are commercially sold (see for example products from company Steico).

Point 6: - What standard did the authors use to conduct flammability tests for? Is this test adequate to the use of these materials in civil engineering? Earlier publication of this flammability testing methodology does not release authors from placing appropriate information in the text of the article.

Response 6: The method used is standardised (EN 1363-2), only slight deviations from the standard were used because the used furnace was custom designed. The flammability test was more deeply described in the methodology part. However, we are still referring to the previous study in order to avoid similarities of the text.

Point 7: -I suggest re-editing "Conclusion" section to a descriptive version. The most important numerical results (that has been improved) should be indicated there and the advantages of this solution should be emphasized.

Response 7: The Conclusions were re-edited according to your suggestions.

Reviewer 2 Report

This manuscript describes the results of research on physico-chemical patameters of sandwich insulation panel consists of a thermal insulation core of recycled soft polyurethane foam and winter wheat husk, a layer of geopolyme and a nanofibrous membrane.

The authors carried out the tests necessary to determine the operational parameters and requirements of this type of materials.

However, the manuscript in its current form cannot be published, because it has many deficiencies. The text requires editorial corrections, add test results and change the manuscript structure, which is now very uncomfortable for the recipient.

In sequence:

Line 84: The picture shows that in both cases the measured thermal insulation cores achieved very good thermal conductivity values at 0.044 W/(m.K).”

On the fig.3 there is other value.

In addition, since it is known that the density is so important for thermal conductivity, why did the authors use polyurethane foam of such density or did not reduce it at the stage of preparing the panel?

Line 108:The influence of nanofiber membranes on thermal insulation properties or fire resistance was not evaluated” Why?

Line 109: “The geopolymer layer only slightly worsened the thermal insulation properties of the sandwich composite.”

I don't see confirmation in the text. With what did the authors compare the thermal insulating properties of geopolymer that draw such conclusions?

Line 144: When the sandwich without a water column was encumbered with a water column with a height of 80 cm, 3700 g of water flowed through the 154 cm2 area over 4 minutes.” Illogical sentence

Figure 7. In the study in which the panel was exposed to a water column 80 cm high, I miss a direct comparison to a panel without a mambrane or reference to other examples from the literature, as the authors did earlier. This would give a better picture of the resulting resistance to water jets.

Line 153: The geopolymer layer was thoroughly bonded to the thermal insulation core, and in the tensile strength test perpendicular to the plane of the board, there was no breach between these layers, but in the insulation core.”

The authors in the text mention, among others here, the tensile strength test perpendicular to the plane of the plate. However, nowhere can I find the results of this study.

Line 159: The "Materials and methods" section should be moved before the “Results and discussion”. In this form, the results are uncomfortably analyzed, because the authors put a lot of real data facilitating interpretations in the next section.

General thoughts:

What was the purpose, idea and benefits of using winter wheat husk as part of the panel core? The authors present 4 panel compositions, and the results represent for a smaller number of selected samples. Maybe it would be worth trying to provide a broader analysis and presentation of research results for a larger number of your samples, or literature reports, because in its current form it is difficult to determine whether the panels designed by the authors have good performance properties, and their further analysis and possibly commercialization would have economic coverage.

Author Response

Dear reviewer,

thank you very much for your valuable suggestions. Please find enclosed to this cover letter the revised manuscript and responses to your comments.

I believe that after carefully performed revisions, the manuscript deserves to be considered for publication in the journal Molecules.

Best regards,

ŠtÄ›pán Hýsek

-----------------------------

Response to Reviewer 2 Comments

Point 1: Line 84: “The picture shows that in both cases the measured thermal insulation cores achieved very good thermal conductivity values at 0.044 W/(m.K).” On the fig.3 there is other value.

In addition, since it is known that the density is so important for thermal conductivity, why did the authors use polyurethane foam of such density or did not reduce it at the stage of preparing the panel?

Response 1: The sentence was corrected.

The aim of this study was to use recycled materials for the production of the panel. The polyurethane foam used is recycled foam from the industry and it was directly used for the production of sandwich panel. The density of the foam is given by the supplier and we assume that any other technological step is needed since the bulk density 11.3 kg/m3 is low enough for the insulation materials.

Point 2: Line 108: „The influence of nanofiber membranes on thermal insulation properties or fire resistance was not evaluated” Why?

Response 2: Because the thin nanofiber membrane does not influence thermal insulation properties and fire resistance.

Point 3: Line 109: “The geopolymer layer only slightly worsened the thermal insulation properties of the sandwich composite.” I don't see confirmation in the text. With what did the authors compare the thermal insulating properties of geopolymer that draw such conclusions?

Response 3: In the table 3 can be seen that λ of the sandwich panel with no husks is 0,049 W/(m.K) and λ of the sandwich panel with husks is 0,051 W/(m.K). Figure 5 shows that λ of the insulation panel without husks is 0,0427 W/(m.K) and λ of the insulation panel with husks is 0,0452 W/(m.K). From these results can be concluded that the geopolymer layer only slightly worsened the thermal insulation properties.

Point 4: Line 144: “When the sandwich without a water column was encumbered with a water column with a height of 80 cm, 3700 g of water flowed through the 154 cm2 area over 4 minutes.” Illogical sentence

Response 4: The sentence was corrected.

Point 5: Figure 7. In the study in which the panel was exposed to a water column 80 cm high, I miss a direct comparison to a panel without a mambrane or reference to other examples from the literature, as the authors did earlier. This would give a better picture of the resulting resistance to water jets.

Response 5: This misunderstanding comes from wrong translation (also connected to Point 4). The panel without a membrane was also tested in this study. The corrected sentence can be find on the line 234.

Point 6: Line 153: “The geopolymer layer was thoroughly bonded to the thermal insulation core, and in the tensile strength test perpendicular to the plane of the board, there was no breach between these layers, but in the insulation core.” The authors in the text mention, among others here, the tensile strength test perpendicular to the plane of the plate. However, nowhere can I find the results of this study.

Response 6: Tensile strength perpendicular to the plane of the board = Internal bonding. We clarified this in the manuscript (Line 118).

Point 7: Line 159: The "Materials and methods" section should be moved before the “Results and discussion”. In this form, the results are uncomfortably analyzed, because the authors put a lot of real data facilitating interpretations in the next section.

Response 7: "Materials and methods" section was moved before the “Results and discussion”. For better clarity, these two chapters were swapped first and then the revision tracking was turned on.

Point 8: What was the purpose, idea and benefits of using winter wheat husk as part of the panel core?

Response 8: In many countries husks after the harvest are burnt on the field. This burning produces annually extreme amounts on CO2 released into the atmosphere. Finding some material utilisation (not utilization for energy purposes as bioethanol or producing of heat) will contribute to storage of the CO2. Selling of these husks may also bring an additional income for farmers.

In this study we confirmed, that addition of 25% of husks only slightly decreases some properties of the panel; and this decrease in absolutely acceptable. What is more, addition of husks may bring some benefits in the production line. The boards made from 100% PUR recyclate and PUR adhesive are currently hardened by steam injection. Since husk contains ca. 12% of moisture, no steam injection would be necessary any more. We are dealing with this problem in our next work.

The explanation was added into the manuscript.

Point 9: The authors present 4 panel compositions, and the results represent for a smaller number of selected samples. Maybe it would be worth trying to provide a broader analysis and presentation of research results for a larger number of your samples, or literature reports, because in its current form it is difficult to determine whether the panels designed by the authors have good performance properties, and their further analysis and possibly commercialization would have economic coverage.

Response 9: We assume that it is not necessary to perform comprehensive and extensive measurements and measure all possible variations experimentally. In our opinion, it is fully sufficient to experimentally verify the boundary possibilities and the results of other variants can be derived or calculated based on results of boundary variants and already known information.

The comparison with other studies and materials was added into the Results and discussion part, the discussion of results was supported by references.

Reviewer 3 Report

Rewision

Foam with adding winter wheat husk are slightly worse conductivity in use with insulation boards made from reeds, bagasse or cotton stalks but they have lower density. Heat insulation board density grows slightly. The influence of nanofiber membranes on thermal insulation properties or fire resistance was not evaluated. There was no difference found between the sandwich panel with the addition of husk and no husk when it comes withstand fairly long-term exposure to a water column with a height of 80 cm.

Points at work are described in the wrong order. The work should be reworded. There are no references in points 2 and 3 (research part).

In the research part of the paper no formulations of recycled flexible foams are presented. The difference in the results of the properties between foams with and without membrane, with and without geopolymer, is not clearly shown, with winter wheat husk and without winter wheat husk. The raw materials for the synthesis of new materials have not been thoroughly discussed (in insulation board). What application does the received material have. Lack of research standards and sample sizes. The purpose of adding wheat husk is not stated. winter wheat husk (is this just a method of managing winter wheat husk waste?). It is not entirely clear in which foams the membrane is used and in which it is not. Werse 19 resistance to long-term exposure of a water column - was the test carried out in accordance with the standard, what standard? The abstract should include improving of the properties of foam with husk. How does geopolymer affect fire resistance. Verse 35 in the introduction says wheat hulls improve insulation. How? What is the cost of producing such a foam with geopolymer? Is it worth it compared to traditional foam? Werse 102 exterior temperatures [31]. Should be temperature, because tere is one temperature. Verse 164-170 no standard is given. Point 3 should be placed before point 2. In item 3 there are no references in the research results. Similarly in item 3.4. Werse 234 Conclusion not conclusions. According to the principles of the journal, conclusion is a uniform text rather than a text in the form of conclusions in points

Author Response

Dear reviewer,

thank you very much for your valuable suggestions. Please find enclosed to this cover letter the revised manuscript and responses to your comments.

I believe that after carefully performed revisions, the manuscript deserves to be considered for publication in the journal Molecules.

Best regards,

ŠtÄ›pán Hýsek

------------------------------------

Response to Reviewer 3 Comments

Point 1: The influence of nanofiber membranes on thermal insulation properties or fire resistance was not evaluated.

Response 1: Because the thin nanofiber membrane does not influence thermal insulation properties and fire resistance.

Point 2: Points at work are described in the wrong order. The work should be reworded. There are no references in points 2 and 3 (research part). Point 3 should be placed before point 2. In item 3 there are no references in the research results. Similarly in item 3.4. Werse 234 Conclusion not conclusions. According to the principles of the journal, conclusion is a uniform text rather than a text in the form of conclusions in points

Response 2: "Materials and methods" section was moved before the “Results and discussion”. For better clarity, these two chapters were swapped first and then the revision tracking was turned on. The discussion of results was supported by references. In the methodology part, the methods are cited by standards or by articles. The Conclusions were re-edited according to your suggestions.

Point 3: In the research part of the paper no formulations of recycled flexible foams are presented. The difference in the results of the properties between foams with and without membrane, with and without geopolymer, is not clearly shown, with winter wheat husk and without winter wheat husk. It is not entirely clear in which foams the membrane is used and in which it is not.

Response 3: We assume that it is not necessary to perform comprehensive and extensive measurements and measure all possible variations experimentally. In our opinion, it is fully sufficient to experimentally verify the boundary possibilities and the results of other variants can be derived or calculated based on results of boundary variants and already known information.

Point 4: The raw materials for the synthesis of new materials have not been thoroughly discussed (in insulation board). What application does the received material have. The purpose of adding wheat husk is not stated. winter wheat husk (is this just a method of managing winter wheat husk waste?).

Response 4: The comparison with other studies and materials was added into the Results and discussion part.

In many countries husks after the harvest are burnt on the field. This burning produces annually extreme amounts on CO2 released into the atmosphere. Finding some material utilisation (not utilization for energy purposes as bioethanol or producing of heat) will contribute to storage of the CO2. Selling of these husks may also bring an additional income for farmers.

In this study we confirmed, that addition of 25% of husks only slightly decreases some properties of the panel; and this decrease in absolutely acceptable. What is more, addition of husks may bring some benefits in the production line. The boards made from 100% PUR recyclate and PUR adhesive are currently hardened by steam injection. Since husk contains ca. 12% of moisture, no steam injection would be necessary any more. We are dealing with this problem in our next work.

The explanation was added into the manuscript.

Point 5: Lack of research standards and sample sizes.

Response 5: All standardised method are cited. The standard for flammability test was added. Sample sizes are defined by the standards. In the case that non-standardised method is used, sample sizes are explicitly listed. Sample sizes for flammability test were added.

Point 6: Werse 19 resistance to long-term exposure of a water column - was the test carried out in accordance with the standard, what standard?

Response 6: This method is not standardised, firstly it was partly presented by our team on the following conference: Fridrichová, L.; Frydrych, M.; Herclík, M.; Knížek, R.; Mayerová, K. Nanofibrous membrane as a moisture barrier. 2018.  In AIP Conference Proceedings. https://doi.org/10.1063/1.5051103

Currently we are preparing paper about his method. The submitted manuscript is the first paper with the method used.

Point 7: The abstract should include improving of the properties of foam with husk. How does geopolymer affect fire resistance.

Response 7: The addition of husk to the thermal insulation core does not significantly impair its thermal insulation properties. The effect of geopolymer layer on the fire resistance is in the abstract reported.

Point 8: Verse 35 in the introduction says wheat hulls improve insulation. How? What is the cost of producing such a foam with geopolymer? Is it worth it compared to traditional foam?

Response 8: These raw materials can be considered for thermal insulation because of air in the structure of the insulation panel from these materials, as is mentioned in the literature cited in the paragraph.

Of course, that in the case that we want to calculate production cost right now, the costs will be high because there is no industrial production of this material. Firstly, it is needed to develop the material and then to develop industrially suitable method of its production.

Point 9: Werse 102 exterior temperatures [31]. Should be temperature, because tere is one temperature.

Response 9: Corrected

Point 10: Verse 164-170 no standard is given.

Response 10: No standard is needed for the board manufacturing. The manufacturing method is described.

Reviewer 4 Report

Dear Authors,

Let me congratulate first.

I would recommend the paper for acceptance after minor revision.

Please do not use straight lines for Figure 3, 4 and 5. I suggest to use dashed lines instead.

Please enlerge the subtitles of figures 3, 4 and 5.

I suggest to calculate the water absorption coefficients from figure 7.

For the method please see: Investigation of the moisture induced degradation of the thermal properties of aerogel blankets: Measurements, calculations, simulations, Energy and Buildings: Volume 139, 15 March 2017, Pages 506-516

THERMAL CHARACTERIZATION OF DIFFERENT GRAPHITE
POLYSTYRENE, Int. Rev. Appl. Sci. Eng. 9 (2018) 2, 163–168
DOI: 10.1556/1848.2018.9.2.12

Author Response

Dear reviewer,

thank you very much for your valuable suggestions. Please find enclosed to this cover letter the revised manuscript and responses to your comments.

I believe that after carefully performed revisions, the manuscript deserves to be considered for publication in the journal Molecules.

Best regards,

ŠtÄ›pán Hýsek

--------------------------------------

Response to Reviewer 4 Comments

Point 1: Please do not use straight lines for Figure 3, 4 and 5. I suggest to use dashed lines instead.

Response 1: The dashed lines were used.

Point 2: Please enlerge the subtitles of figures 3, 4 and 5.

Response 2: The subtitles were enlarged.

Point 3: I suggest to calculate the water absorption coefficients from figure 7.

For the method please see: Investigation of the moisture induced degradation of the thermal properties of aerogel blankets: Measurements, calculations, simulations, Energy and Buildings: Volume 139, 15 March 2017, Pages 506-516

THERMAL CHARACTERIZATION OF DIFFERENT GRAPHITE

POLYSTYRENE, Int. Rev. Appl. Sci. Eng. 9 (2018) 2, 163–168

DOI: 10.1556/1848.2018.9.2.12

Response 3: From our data it is not possible to calculate the water absorption coefficients according the methodology in the papers, because our panels were exposed to the water column of 0.8 m height. We did not record amount of water in the panel, but amount of water that went through the panel.

Round 2

Reviewer 1 Report

Article entitled "Permeable Water-Resistant Heat Insulation Panel Based on Recycled Materials and its Physical and Mechanical Properties" has been significantly improved. It is more readable than the original manuscript. Most of my comments have been taken into account. However, I have a few more comments about it:

1) Answer to point 5. Authors wrote that the insulations offered by Steico have similar λ I agree, but these are completely different types of insulation than those obtained by the authors. If authors use polyurethane foam as a core of panel, they should compare the obtained results with λ coefficients for commercial PUR or PIR panels.

2) Line 80 – apparent density of foam not density.

3) Some sentences in the text need to be corrected, e.g.

“These results correspond with [13] and because of flammable insulation core the panel withstood lower temperature than in [16].”

Authors should write, e.g.

“These results correspond with result for sandwich-structured composites modified by winter rape stalks [13]…” etc.

These types of lacks must be completed. This will make it easier for a potential reader to read this article, without having to check the cited references every time.

4) Many references do not contain a DOI number. The authors should add this.

Author Response

Dear reviewer,

thank you for your valuable comments:

Point 1: Answer to point 5. Authors wrote that the insulations offered by Steico have similar λ I agree, but these are completely different types of insulation than those obtained by the authors. If authors use polyurethane foam as a core of panel, they should compare the obtained results with λ coefficients for commercial PUR or PIR panels.

Response 1: The comparison was added:

The reached thermal conductivity coefficients are higher, that thermal conductivity coefficients of commercially produced heat insulation panels from PUR or PIR (polyisocyanurate) panels, however, the developed panels are from recycled materials and from recycled PUR that was initially not produced for thermal insulation.

Point 2: Line 80 – apparent density of foam not density.

Response 2: Corrected

Point 3: Some sentences in the text need to be corrected, e.g.

“These results correspond with [13] and because of flammable insulation core the panel withstood lower temperature than in [16].”

Authors should write, e.g.

“These results correspond with result for sandwich-structured composites modified by winter rape stalks [13]…” etc.

These types of lacks must be completed. This will make it easier for a potential reader to read this article, without having to check the cited references every time.

Response 3: The sentence was reformulated.

Point 4: Many references do not contain a DOI number. The authors should add this.

Response 4: The DOI numbers were added.

Best regards

Stepan Hysek

Reviewer 2 Report

The manusktupt regarding evelopment and characteristics of the properties of a permeable water-resistant heat insulation panel based on recycled materials is now available for publication.

Author Response

Thank you for your positive review!

Reviewer 3 Report

The work was significantly improved, in line with the magazine's comments and requirements.
Can be accepted for printing.

Author Response

Thank you for your positive review!